# Anticonvulsant Effect of Turmeric and Resveratrol in Lithium/Pilocarpine-Induced Status Epilepticus in Wistar Rats

**DOI:** 10.3390/molecules27123835

**Published:** 2022-06-14

**Authors:** Isaac Zamora-Bello, Eduardo Rivadeneyra-Domínguez, Juan Francisco Rodríguez-Landa

**Affiliations:** 1Facultad de Química Farmacéutica Biológica, Universidad Veracruzana, Xalapa 91000, Mexico; iszamora@uv.mx (I.Z.-B.); juarodriguez@uv.mx (J.F.R.-L.); 2Laboratorio de Neurofarmacología, Instituto de Neuroetología, Universidad Veracruzana, Xalapa 91190, Mexico

**Keywords:** epilepsy, turmeric, resveratrol, blood cytometry, liver function, renal function

## Abstract

Epilepsy is a chronic neurological disorder that lacks a cure. The use of plant-derived antioxidant molecules such as those contained in turmeric powder and resveratrol may produce short-term anticonvulsant effects. A total of 42 three-month-old male Wistar rats were divided into six groups (*n* = 7 in each group): Vehicle (purified water), turmeric (150 and 300 mg/kg, respectively), and resveratrol (30 and 60 mg/kg, respectively), administered *per os* (p.o.) every 24 h for 35 days. Carbamazepine (300 mg/kg/5 days) was used as a pharmacological control for anticonvulsant activity. At the end of the treatment, status epilepticus was induced using the lithium–pilocarpine model [3 mEq/kg, intraperitoneally (i.p.) and 30 mg/kg subcutaneously (s.c.), respectively]. Seizures were evaluated using the Racine scale. The 300 mg/kg of turmeric and 60 mg/kg of resveratrol groups had an increased latency to the first generalized seizure. The groups treated with 150 and 300 mg/kg of turmeric and 60 mg/kg of resveratrol also had an increased latency to status epilepticus and a decreased number of generalized seizures compared to the vehicle group. The chronic administration of turmeric and resveratrol exerts anticonvulsant effects without producing kidney or liver damage. This suggests that both of these natural products of plant origin could work as adjuvants in the treatment of epilepsy.

## 1. Introduction

Epilepsy is a disorder of the central nervous system (CNS), characterized by abnormal and synchronized electrical activity of neurons in a specific area of the brain [1,2,3]. Status epilepticus (SE) is a type of epileptic activity that includes two or more sequential seizures with an average duration of 30 min, without recovery of consciousness between seizures [4]. SE is successively related to epileptogenesis and neuronal damage and death, both in humans and animals [5,6]. The treatment of epilepsy mainly includes drugs such as valproic acid, carbamazepine, phenytoin, gabapentin, and topiramate [7]. These are effective in most cases; however, some patients discontinue the use of anticonvulsant drugs due to their undesirable effects, while other patients are unresponsive to conventional treatments [7]. In this sense, patients seek treatment alternatives such as the use of plants that have apparent medicinal properties, particularly those with neuroprotective properties and biological effects such as antioxidant, anti-inflammatory, and anticancer effects. Such properties are produced by molecules of diverse chemical natures (secondary metabolites), which vary according to the plant species [8]. A considerable number of plants contain molecules such as flavonoids, polyphenols, glycosides, tannins, and triterpenes with marked antioxidant activity, which are used as antispasmodics, antithrombotics, contraceptives, hypoglycemic agents, and hepatoprotectors. However, it is noteworthy that these substances also exert effects on the CNS and are used to treat patients with Alzheimer’s disease (AD) and epilepsy [8]. *Curcuma longa*, commonly called turmeric, is an edible plant of the Zingiberaceae family, whose medicinal properties are associated with the content of curcuminoids [8]. On the contrary, resveratrol is a polyphenolic flavonoid contained in the skin of grapes, blueberries, and blackberries. Its consumption has health benefits, such as its antiglycemic effects, decreased platelet aggregation, and ability to decrease the harmful effects of a high-fat diet [9,10]. Both turmeric and resveratrol have antioxidant properties and could be potential neuroprotectors against neurodegenerative diseases where toxicity processes associated with free radicals are involved. In this sense, a neuroprotective effect of both of the molecules has been observed in Parkinson’s disease (PD), AD, and brain trauma, all related to their antioxidant properties [11,12,13]. During SE seizures, free radicals are generated and contribute to glutamate-facilitated cytotoxicity, which induces apoptosis. The greater the severity and duration of the seizures, the greater the cell damage. The aim of this study was to evaluate the effect of long-term treatment with turmeric and resveratrol on epileptic seizures induced by lithium/pilocarpine in rats. In addition, renal and liver function and blood chemistry were evaluated to identify potential toxic effects that could harm the individual. The contribution of this study is an evaluation of the antiepileptic effects after long-term treatment with turmeric and resveratrol, as well as potential hepatic and renal damage. This could contribute to the development of complementary therapeutic strategies for the treatment of epilepsy and the attenuation of its long-term neurological effects.

## 2. Results

### 2.1. Evaluation of Status Epilepticus

The carbamazepine group was devoid of phase IV or V generalized seizures and SE; however, three individuals in this group developed head myoclonus without generalized seizures or SE. Therefore, this group was only included in the statistical analysis of the biochemical and hematological parameters.

Analysis of the latency to the first generalized seizures (phase IV or V) revealed significant differences between the groups (*F*(4, 30) = 14.7, *p* < 0.0001). The 300 mg/kg of turmeric and 60 mg/kg of resveratrol-treated groups had a significantly higher latency compared to the vehicle group. The 150 mg/kg of turmeric and 30 mg/kg of resveratrol groups were devoid of effects on this variable (Figure 1).

The latency to SE revealed significant differences between the groups (*F*(4, 30) = 19, *p* < 0.0001). This variable was higher in the 150 and 300 mg/kg of turmeric and 60 mg/kg of resveratrol groups compared to the vehicle group. The 30 mg/kg of resveratrol group was devoid of effects on this variable (Figure 2).

Regarding the duration of the first generalized seizures (phase IV or V), significant differences between groups (*F*(4, 30) = 9.1, *p* = 0.0001) were also revealed. The post-hoc test revealed that the 300 mg/kg of turmeric group produced a shorter time compared to the vehicle group. The 150 mg/kg of turmeric and 30 and 60 mg/kg of resveratrol groups were devoid of effects on this variable (Figure 3).

Analysis of the number of generalized phase IV seizures did not reveal any significant differences between the groups (*F*(4, 30) = 2.4, *p* = 0.068). However, analysis of the number of generalized phase V seizures revealed significant differences between the groups (*F*(4, 30) = 5.6, *p* = 0.0018). The post-hoc test indicated that the 150 and 300 mg/kg of turmeric and 60 mg/kg of resveratrol groups had a reduced number of generalized phase V seizures compared to the vehicle group. The 30 mg/kg of resveratrol group was devoid of effects on this variable (Figure 4).

### 2.2. Renal Function Tests

Analysis of the parameters that made up the blood chemistry did not reveal any significant differences between the treatment groups; the values were within the reference intervals: Glucose (*F*(5, 36) = 1.47, *p* = 0.134), urea (*F*(5, 36) = 1.77, *p* = 0.143), creatinine (*F*(5, 36) = 1.44, *p* = 0.233), and Blood Urea Nitrogen (BUN) (*F*(5, 36) = 1.92, *p* = 0.114) (Table 1).

### 2.3. Hepatic Function Tests

The parameters that made up the liver function tests were not affected by the treatments. The results were found to be within the reference intervals: Total bilirubin (*F*(5, 36) = 1.97, *p* = 0.106), direct bilirubin (*F*(5, 36) = 1.87, *p* = 0.335), indirect bilirubin (*F*(5, 36) = 1.98, *p* = 0.104), aspartate aminotransferase (AST) (*F*(5, 36) = 7.39, *p* = 0.140), alanine aminotransferase (ALT) (*F*(5, 36) = 3.75, *p* = 0.091), alkaline phosphatase (*F*(5, 36) = 3.09, *p* = 0.020), total protein (*F*(5, 36) = 1.94, *p* = 0.110), albumin (*F*(5, 36) = 2.47, *p* = 0.082), and γ-glutamyl-transferase (*F*(5, 36) = 1.4, *p* = 0.22) (Table 2).

### 2.4. Blood Cytometry

Analysis of the parameters that made up the hematic cytometry did not reveal any significant differences between the groups; the results were within the reference intervals: Erythrocytes (*F*(5, 36) = 0.61, *p* = 0.688), hemoglobin (Hb) (*F*(5, 36) = 2.36, *p* = 0.059, hematocrit (Hto) (*F*(5, 36) = 0.69, *p* = 0.633), Mean Corpuscular Volume (MCV) (*F*(5, 36) = 0.32, *p* = 0.894), Mean Cell Hemoglobin (MCH) (*F*(5, 36) = 2.87, *p* = 0.278), Mean Corpuscular Hemoglobin (MCHC) (*F*(5, 36) = 2.48, *p* = 0.938), leukocytes (*F*(5, 36) = 1.81, *p* = 0.135), and platelets (*F*(5, 36) = 1.01, *p* = 0.422) (Table 3).

### 2.5. Blood Count

The differential leukocyte count did not reveal any significant differences between the groups; all of the values were within the reference intervals: Segmented (mature) neutrophils (*F*(5, 36) = 1.44, *p* = 0.232), band (immature) neutrophils (*F*(5, 36) = 1.77, *p* = 0.142), lymphocytes (*F*(5, 36) = 2.62, *p* = 0.402), eosinophils (*F*(5, 36) = 0.677, *p* = 0.643), basophils (*F*(5, 36) = 1.69, *p* = 0.160), and monocytes (*F*(5, 36) = 1.99, *p* = 0.142) (Table 4).

## 3. Discussion

The aim of the present study was to evaluate the anticonvulsant effect of the long-term oral administration of different doses of turmeric and resveratrol and their impact on the biochemical and hematological parameters in rats with SE induced by lithium–pilocarpine. The results are summarized as follows: (a) Chronic administration of turmeric and resveratrol showed an anticonvulsant effect against SE, which is supported by the long latency of onset of generalized epileptic activity and SE, in addition to the decreased number of generalized phase V seizures in the 150 and 300 mg/kg of turmeric and 60 mg/kg of resveratrol groups. (b) Chronic administration of turmeric and resveratrol showed no effects in the renal and hepatic function tests, or in the blood cytometry. The foregoing indicates that both compounds could be potential therapeutic alternatives for the control of SE, without generating adverse effects in the organism.

In the present study, carbamazepine prevented the development of generalized phase IV or V seizures and SE, for which it was not possible to evaluate the behavioral parameters. Consistent with this, carbamazepine is a drug commonly used for the treatment of epilepsy; it is useful in the control of generalized tonic–clonic seizures, which have also been induced in the SE of the lithium–pilocarpine model [20]. Carbamazepine (100 mg/kg) completely blocked kainate-induced seizures in 70–89% of rats [21]. These data pharmacologically validate the model used in this study.

The administration of turmeric and resveratrol modified the behavior of SE in the following aspects: An increased latency to the first generalized seizure and to SE and a decreased duration of the first generalized seizure and the number of phase V generalized seizures. The above shows that the administration of turmeric and resveratrol decreases the severity of seizures and delays their appearance. In this way, the dose–response, which involves the principles of pharmacokinetics and pharmacodynamics, determines the dose and/or concentration and the therapeutic index of a drug in a population. However, increasing the dose of a drug does not, in all cases, lead to a greater effect (for example, increasing a drug with a small therapeutic range increases the risk of it becoming ineffective or toxic). Regardless of the mechanism by which a drug exerts its effects, be it by receptor binding or chemical interaction, control of the effect depends on the concentration of the drug at the site of action. In short, the receptors with the low dose could be saturated, and in this way, a ceiling effect is presented, in which, despite the fact that the dose is increased, the side effects can be increased without these being translated into greater efficacy. In this study, the groups that had a higher dose of resveratrol presented a greater number of seizures, or the groups that had a higher dose of turmeric did not necessarily present a longer latency in this study.

Previous studies have reported the anticonvulsant effects of turmeric. In an epileptogenic model of pentylenetetrazole (PTZ; 30 mg/kg/day) in rats, turmeric (300 mg/kg p.o.) was administered daily for 31 ± 1 days, after which an anticonvulsant effect was shown by a significant increase in the latency to myoclonic jerks, clonic seizures, and generalized tonic–clonic seizures, in addition to decreasing the duration of generalized tonic–clonic seizures [22]. In the model of epileptogenesis generated by a unilateral intrahippocampal injection of 4 μg of kainate, rats that received 100 mg/kg/day of curcumin, the active ingredient in turmeric, had a reduction in seizures in 71.5% of the individuals [23]. In addition, it prevented a decrease in the number of *Cornu Ammonis* (CA1) pyramidal cells associated with seizures [18]. The studies described above used chronic models of epilepsy generation (epileptogenesis), a dynamic process by which the brain begins to generate spontaneous and recurrent epileptic seizures [22,23]—while our study used SE. In SE, the generation of seizures are different and produced through acute administration, can be systemic or intracerebral of convulsive agents (pilocarpine generally) [24].

Resveratrol also reduces the severity of seizures in rats. Intragastric administration of resveratrol 15 mg/kg of PTZ decreased the frequency of spontaneous seizures and inhibited epileptiform discharge induced by kainic acid. Resveratrol administration started after the animals showed generalized seizures of phase IV on the Racine scale [25]. In addition, the administration of 20 mg/kg of resveratrol in male Sprague–Dawley rats decreased the peak frequency and amplitude of seizure activity induced by penicillin [26]. Resveratrol (40 mg/kg) administered 20 min before the convulsant PTZ (60 mg/kg i.p.) reduced generalized tonic–clonic seizures. Furthermore, it potentiated the effect of sodium valproate (150 mg/kg) and diazepam (2 mg/kg) in inhibiting PTZ-induced seizures [27]. The studies describing the influence of resveratrol on the severity of seizures show similarities and differences to our study: Kainic acid is an analog of glutamate, the chronic administration of which can induce the process of epileptogenesis and seizures originating in the limbic system [25]. Meanwhile, in our study lithium–pilocarpine was used to induce SE, which is an acute model in Wistar rats. Pilocarpine has cholinergic action, acting at the level of Muscarinic Receptor 1 (M1). In another study, penicillin was used as a convulsant, generating a cortical epileptic focus in rats of the Sprague–Dawley strain [26]. For its part, PTZ is a non-competitive antagonist of GABAA (Gamma Amino Butyric Acid) receptors that generates tonic–clonic seizures administered acutely in Wistar rats [27]. Due to the aforementioned, each of the models has a different mechanism of action. One of the advantages of the lithium–pilocarpine model is that, being an acute model, it is cheaper compared to an epileptogenesis model; in addition to this, it mimics temporal lobe epilepsy, which is the most frequent in humans. Our results, in agreement with previous reports, show the potential of turmeric and resveratrol as adjuvants in the reduction in behavioral manifestations of epilepsy.

On the contrary, blood chemistry is used to determine the concentrations of different substances present in the blood. The most used parameters to evaluate renal function are glucose, BUN, urea, and creatinine [28]. The results of the present study reveal that these parameters were unaffected, evidencing the safety of the prolonged consumption of turmeric and resveratrol against adverse reactions or changes in renal function under our experimental conditions. Regarding chronic treatment with carbamazepine, there are no reports that identify or rule out potential kidney damage, but there are data on its toxicity at the hepatic level.

Hepatic function tests measure the presence of some enzymes, proteins, and degradation products from the liver [29]. In this study, the treatments were devoid of effects on the aforementioned parameters, which shows the safety of the prolonged consumption of turmeric and resveratrol against adverse reactions or hepatotoxicity. There are reports of toxicity of carbamazepine in rodents. Consecutive five-day administration of oral 400 mg/kg of Carbamazepine (CBZ) for four days and 800 mg/kg on the fifth day increased the plasma levels of ALT, AST, and centrilobular necrosis in the liver of male BALB/c mice [30]. However, a dose of 400 mg/kg for five consecutive days showed no effects on the plasma levels of ALT and AST [30]. Consistent with this, our results show that 300 mg/kg of carbamazepine every 48 h for five alternate days does not alter the plasma levels of ALT and AST.

Blood cytometry is a clinical laboratory test that is integrated with the red, white, and hemostasis formula, which allows the diagnosis of hematological pathologies of different organs and systems [31]. In the present study, all of the blood cytometry values were found within the reference intervals, and, with regard to the observation under the microscope in immersion of the blood smears of each of the samples belonging to the individuals of the different groups, we also found that they were within the reference intervals. Likewise, no morphological alterations in the number of cells were identified, so this study also rules out the possible toxicity of turmeric and resveratrol that could alter erythropoiesis, leukopoiesis, and thrombopoiesis under these experimental conditions.

Regarding the way resveratrol and turmeric cross the blood–brain barrier (BBB), it is known that once incorporated into the diet, they can have several effects; one of them is as an antioxidant. This antioxidant capacity promotes the ability of the cell to cancel reactivity and/or inhibit the generation of free radicals and thus limits diseases related to oxidative stress [32]. In this sense, once resveratrol is administered orally in rats, it is rapidly absorbed and metabolized by glucuronides or sulfate conjugates, mainly by phase II enzymes, and is distributed to various organs [33,34]. At the intestinal level, resveratrol is absorbed, in its glycosylated form, either by passive diffusion or by forming complexes with membrane transporters, reaching the blood 30 min after oral intake [35]. Once in the blood, resveratrol can be transported bound to albumin and lipoproteins such as LDL (low-density lipoproteins); the complex formed can be dissociated in cell membranes that contain receptors for albumin and LDL, leaving resveratrol free and allowing it to penetrate cells [36]. Resveratrol passes into the blood circulation after the formation of glucuronide conjugates, subsequently crossing the BBB [9]. Turner and colleagues [37] showed that the main metabolite found in the rat brain after resveratrol consumption is resveratrol-3-glucuronic acid (cis form), which was also found in the plasma. Therefore, the antioxidant properties of resveratrol are used to protect against neuronal damage in neurodegenerative disorders [32]. On the contrary, curcumin is the most abundant polyphenol in *Curcuma longa*, to which its medicinal and antioxidant properties are attributed [11]. The lipophilicity of turmeric allows gastrointestinal absorption by passive diffusion. In healthy people, the consumption of 10–12 g of curcumin has been detected free in plasma one hour after ingestion [38]. In the small intestine, curcumin undergoes microbial metabolism and a medium chain enzyme of the dehydrogenase/reductase superfamily (curcumin/dihydrocurcumin reductase NADPH-dependent) metabolizes it in two reduction stages. It is first converted (NADPH-dependently) to the intermediate product dihydrocurcumin and then to the final product tetrahydrocurcumin [39]. Due to its lipophilic nature, curcumin (the active ingredient in turmeric) crosses the blood–brain barrier [40]. Therefore, the antioxidant properties of turmeric [41] could be used to protect against neuronal damage in neurodegenerative disorders.

Finally, although this study did not explore the mechanism by which chronic oral administration of turmeric and resveratrol produces an anticonvulsant effect against lithium–pilocarpine-induced SE, it is suggested that these effects could be related to protection against possible damage that generates seizures. In this sense, pilocarpine reaches the CNS and acts as an agonist of the muscarinic receptors M1 and M2, both coupled to Gq proteins [42]. Activation of the M1 receptor has convulsive effects [43,44]. The M1 receptor is located on pyramidal cells in the CA1 and CA3 regions of the hippocampus and in some interneurons. In CA1, the M1 receptor colocalizes with N-Methyl-D-Aspartate (NMDA) receptors [45]. When pilocarpine binds to the M1 receptor, phospholipase C is activated, producing inositol triphosphate (IP3) and diacylglycerol (DAG). IP3 promotes the release of intracellular Ca++ from internal stores, mainly from the endoplasmic reticulum. The intracellular increase in calcium promotes the release of glutamate, which produces seizures and induces SE [46].

It is likely that the anticonvulsant effect of resveratrol involves the modulation of GABAA and SIRT1 receptors (Sirtuin expression activator 1), considering that it reduces generalized tonic–clonic seizures and potentiates the effect of two GABAergic drugs, sodium valproate and diazepam, against PTZ-induced seizures [27]. Furthermore, resveratrol is a modulator of GABAA-induced currents and reduces neuroinflammation in AD patients [47]. It also potentiates GABAA and GABAB receptor-mediated inhibitory postsynaptic currents (IPSCs) in the neurons of the ventral tegmental area [48]. In addition, resveratrol activates SIRT1 (a protein expressed in the cortex, hippocampus, cerebellum, and hypothalamus) under oxidative stress, which regulates the cognitive function in mice by maintaining synaptic plasticity [49]. SIRT1 has protective effects against CNS diseases, including cerebral ischemia, Huntington’s disease, AD, and Parkinson’s disease [49,50]. On the contrary, the antioxidant properties of turmeric [44] could be responsible for its anticonvulsant effects. The exact mechanism is still unclear. It is suggested that it may involve the modulation of GABA and the expression of SIRT1. Curcumin improves the metabolism of GABA in the hypothalamus of rats, increasing its levels. In addition, it can inhibit GABA transaminase or succinic semialdehyde dehydrogenase (major enzymes in GABA catabolism) [51]. SIRT1 expression and activity have been found to be restored in the hippocampal CA3 region of rats after kainic acid-induced status epilepticus, and increase the peroxisomal proliferator-activated receptor α (PGC-1α)/mitochondrial antioxidant coactivator signaling pathways [52]. All of these effects, at the cellular level, could contribute to the neuronal mechanisms underlying the anticonvulsant effects of turmeric and resveratrol, which should be explored in further studies (Figure 5).

## 4. Materials and Methods

### 4.1. Ethical Approval

The experimental protocols were performed strictly according to the Guide for the Care and Use of Laboratory Animals [53] and Official Mexican Standard NOM-062-ZOO-1999 [54]; additionally, the recommendations stated by the three Rs of Russell (reduce, replace, and refine), as applied to experimental research in animals, were considered [55]. The protocol was approved by the Internal Committee for the Care and Use of Laboratory Animals of the Institute of Health Sciences (CICUAL-ICS) with registration number 2018-002B, dated 18 February 2019.

### 4.2. Animals

This study included 42 adult male Wistar rats, weighing 250–300 g at the beginning of the experiments. The rats were housed in Plexiglas cages (four rats per cage) under a 12 h/12 h light/dark cycle (lights on at 07:00 a.m.) and an average room temperature of 25 ± 2 °C. The animals had ad libitum access to water and food.

### 4.3. Dose Selection

The resveratrol doses were 30 and 60 mg/kg/day, respectively, according to Wang et al. [56], who found a neuroprotective effect with both doses. In the case of turmeric, the doses used were 150 and 300 mg/kg/day, respectively, which produced a neuroprotective effect [57]. Carbamazepine was used as a pharmacological control, considering that it produces anticonvulsant activity [21,58]. Turmeric was purchased in a food supplement formulation under the trade name Turmeric^©^ (General Nutrition Centers GNC Laboratories, Pittsburgh, PA, USA; each capsule weighed 0.06 g, containing 350 mg of standardized turmeric). Calculation for 150 mg/kg of turmeric: Protected from light, the content of capsules with 350 mg of turmeric was diluted in 2.3 mL of purified water to obtain a concentration of 150 mg/mL of working solution (note: 1 mL of the working solution was used for 1 kg of body weight of the animal, and a dose of 150 mg/kg was obtained). Next, 1 mL of the working solution was taken and, by the rule of three, the volume required for a rat weighing 300 mg was obtained, which was equal to 0.3 mL of the working solution. Calculation for 300 mg/of turmeric: Protected from light, the content of capsules with 350 mg of turmeric was diluted in 1.167 mL of purified water to obtain a concentration of 300 mg/mL of working solution (note: 1 mL of the working solution was used for 1 kg of animal weight, obtaining a dose of 300 mg/kg). Then, 1 mL of the working solution was taken and, by the rule of three, the volume required for a rat weighing 300 mg was obtained, which was equal to 0.3 mL of the working solution.

Resveratrol was purchased under the trade name of ResVitále^©^ (GNC Laboratories; each capsule weight 1.1 g, containing 500 mg of resveratrol). For this research, we used products with a pharmaceutical form and a standardized dose per unit, without reagent-grade active ingredients, because the civilian population has access to commercial products and consumes them as food supplements. Calculation for 30 mg/kg of resveratrol: Protected from light, the content of capsules with 500 mg of resveratrol was diluted in 16.67 mL of purified water to obtain a concentration of 30 mg/mL of the working solution (note: 1 mL of the working solution was used for 1 kg of animal weight, obtaining a dose of 30 mg/kg). Next, 1 mL of the working solution was taken and, by the rule of three, the volume required for a rat weighing 300 mg was obtained, which was equal to 0.3 mL of the working solution. Calculation for 60 mg/kg of resveratrol: Protected from light, the content of capsules with 500 mg of resveratrol was diluted in 8.34 mL of purified water to obtain a concentration of 60 mg/mL of the working solution (note: 1 mL of the working solution was used for 1 kg of animal weight, obtaining a dose of 60 mg/kg). Then, 1 mL of the working solution was taken and, by rule of three, the volume required for a rat weighing 300 mg was obtained, which was equal to 0.3 mL of the working solution.

For oral administration, a rounded-tip polyethylene intragastric cannula (4 cm long, 1 mm in diameter; S-54-HLCole-Parmer, VernonHills, IL, USA) was used. The animal was held in an upright position and the cannula was passed through one side of the oral cavity, between the incisors and premolars, into the esophagus, where the turmeric or resveratrol solutions were deposited [16].

### 4.4. Experimental Groups

A total of 42 rats were randomly assigned to six independent groups (*n* = 7 rats/group) using free software available online (https://random.org, (accessed on 14 October, 2019)). Groups: Vehicle (purified water), 30 mg/kg of resveratrol, 60 mg/kg of resveratrol, 150 mg/kg of turmeric, 300 mg/kg of turmeric, and 300 mg/kg of carbamazepine. Each dose was prepared in a volume of 2 mL of purified water as the vehicle. The treatment was administered every 24 h for 35 consecutive days, with the exception of carbamazepine, which was administered for five days to prevent liver damage and was used as a pharmacological control for the anticonvulsant activity of SE [21,59]. On the day after the last administration of the treatments, all of the experimental subjects underwent SE induction by lithium–pilocarpine and behavioral assessment (described in detail below). Once the behavioral test was completed, the animals were subjected to deep anesthesia induced with sodium pentobarbital (60 mg/kg, i.p.; Cheminova de México, Mexico City, Mexico; Reg. SAGARPA Q-7048-044) to reach a surgical level (an unconscious state and free of pain and distress, with the absence of palpebral reflex responses and of paw and tail pad pinching). Cardiac puncture was performed using a 5-mL syringe with a 22 G × 32 mm needle by inserting it from the left side of the chest between the third and fifth ribs at the point of maximum identification of heart palpitation. The needle was then moved slowly, thus making slight negative pressure in the cylinder of the syringe, and then we carefully drew blood until blood flow stopped. The blood was deposited into dry tubes (without anticoagulant) with ethylenediaminetetraacetic acid (EDTA; BD Vacutainer, Mexico City, Mexico) anticoagulant and processed and stored until biochemical and hematological analysis.

### 4.5. Induction of Status Epilepticus with Lithium–Pilocarpine

On the last day of turmeric or resveratrol administration (treatment day 35), the animals were administered lithium chloride (3 mEq/kg, i.p. equivalent to a dose of 127.2 mg/kg; Meyer Reagents. Mexico City, Mexico), and 20 h later, pilocarpine hydrochloride (30 mg/kg, s.c.; Sigma-Aldrich. St. Louis, MO, USA) was administered to induce SE. The test sessions were videotaped, and the behavioral manifestations of seizures were recorded according to the Racine scale [59,60]: Phase 0, behavioral arrest; phase I, facial myoclonus (wink ipsilateral to the site of stimulation and/or chewing); phase II, behaviors of phase I and head myoclonus (repeated downward head tilt and nodding); phase III, behaviors of phase II and myoclonus of the forelimbs; phase IV, all of the previous behaviors and kangaroo posture; phase V, behaviors of phase IV and the loss of postural tone and falling. When the seizure activity was continuous or intermittent and persisted for at least 30 min, SE was considered. Only those animals that presented continuous seizures for a period greater than 30 min with a severity of at least stage IV or V (Racine) [60] were included in the behavioral seizure registry. The parameters evaluated included: (1) Latency to first generalized seizure (stage IV or V) after pilocarpine injection, (2) latency to SE after pilocarpine injection, (3) duration of first generalized seizure, and (4) the number of phase IV and phase V generalized seizures.

### 4.6. Blood Samples

The blood samples from tubes without EDTA containing blood were allowed to clot. Subsequently, they were centrifuged at 3500 revolutions per minute for 5 min to obtain serum, which was immediately transferred with a Pasteur pipette to the corresponding containers to perform the indicated tests in dry chemistry in the Vitros 250 equipment (Johnson and Johnson, Ramsey, MN, USA). For the blood cytometry, the samples from the EDTA tubes were homogenized by immersion and subsequently processed in an automatic closed mode on the Advia 560 analyzer (Siemens Healthcare Diagnostics Inc. Tarrytown, NY, USA).

The results were compared with the reference intervals corresponding to Wistar rats, which the equipment itself has programmed, in order to interpret the data and identify possible alterations in the blood cytometry and kidney and liver function tests.

### 4.7. Statistical Analysis

For the statistical analysis, the GraphPad Prism version 6c program was used, considering a significance level of *p* < 0.05. One-way ANOVA was used, considering treatment as the only factor. Tukey’s test was used as a post-hoc test. The results are presented as the mean ± SEM.

## 5. Conclusions

In conclusion, the oral administration of turmeric and resveratrol for 35 consecutive days exerts anticonvulsant effects without causing kidney or liver damage or alterations in hematic cytometry. This suggests that both substances could serve as adjuvants in the treatment of epilepsy, with a certain degree of safety. However, this possibility needs to be explored in depth before being evaluated in human studies.

## Figures and Tables

**Figure 1 molecules-27-03835-f001:**
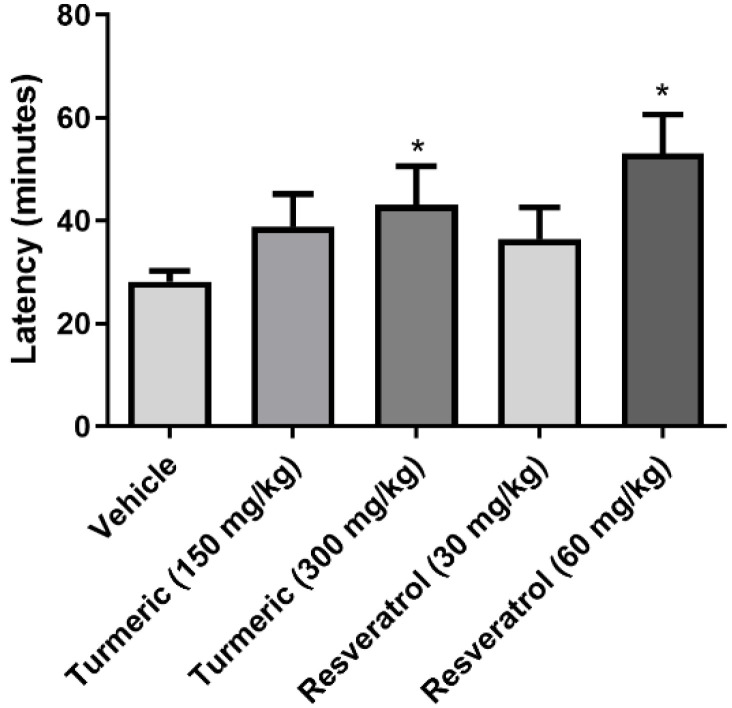
Latency to first generalized seizure (IV/V) after pilocarpine injection. The 300 mg/kg of turmeric and 60 mg/kg of resveratrol groups prolonged the latency to the first generalized seizure compared to the vehicle group. Data were analyzed by one-way ANOVA and Tukey’s post-hoc test. Data are expressed as the mean ± standard error. * *p* < 0.05 vs. the vehicle group.

**Figure 2 molecules-27-03835-f002:**
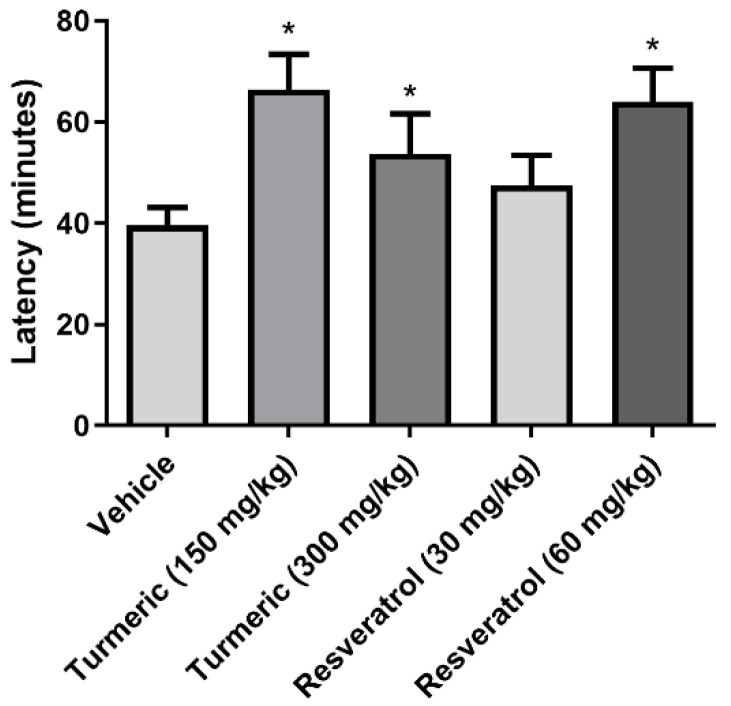
SE latency after pilocarpine injection. The 150 and 300 mg/kg of turmeric and 60 mg/kg of resveratrol groups prolonged SE latency compared to the vehicle group. Data were analyzed by one-way ANOVA, followed by Tukey’s multiple comparison test. Data are expressed as the mean ± standard error. * *p* < 0.05 vs. the vehicle group.

**Figure 3 molecules-27-03835-f003:**
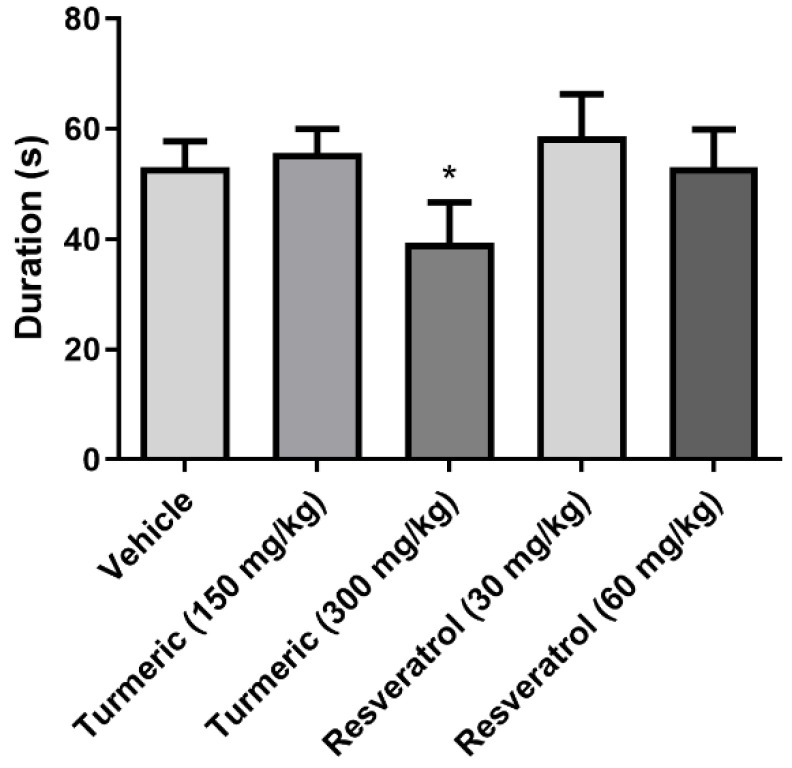
Duration of the first generalized seizure. The 300 mg/kg of turmeric group decreased the duration of the first generalized seizure compared to the vehicle group. Data were analyzed by one-way ANOVA one-way, followed by Tukey’s multiple comparison test. Data are expressed as the mean ± standard error. * *p* < 0.05 vs. the vehicle group.

**Figure 4 molecules-27-03835-f004:**
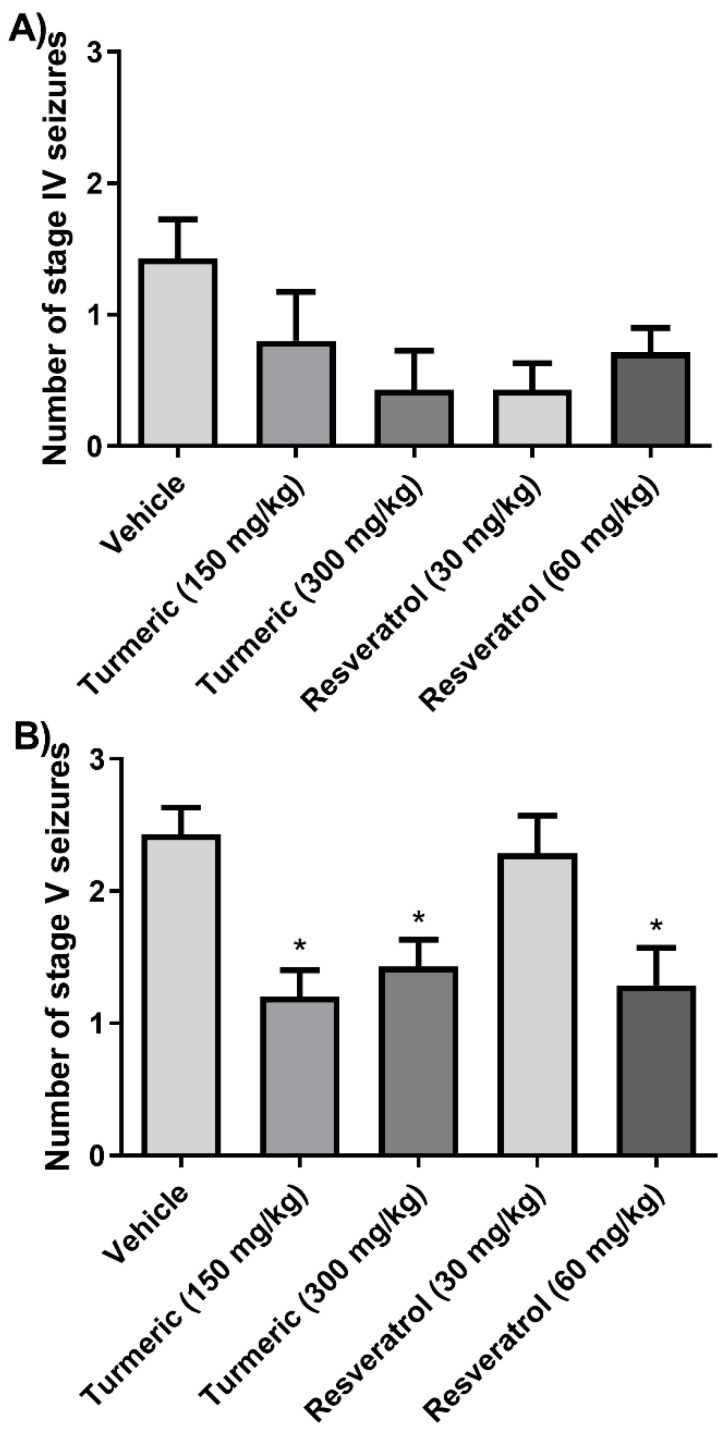
Number of (**A**) phase IV and (**B**) phase V generalized seizures. The 150 and 300 mg/kg of turmeric and 60 mg/kg of resveratrol groups decreased the number of phase V generalized seizures compared to the vehicle group. Data were analyzed by one-way ANOVA, followed by Tukey’s multiple comparison test. Data are expressed as the mean ± standard error. * *p* < 0.05 vs. the vehicle group.

**Figure 5 molecules-27-03835-f005:**
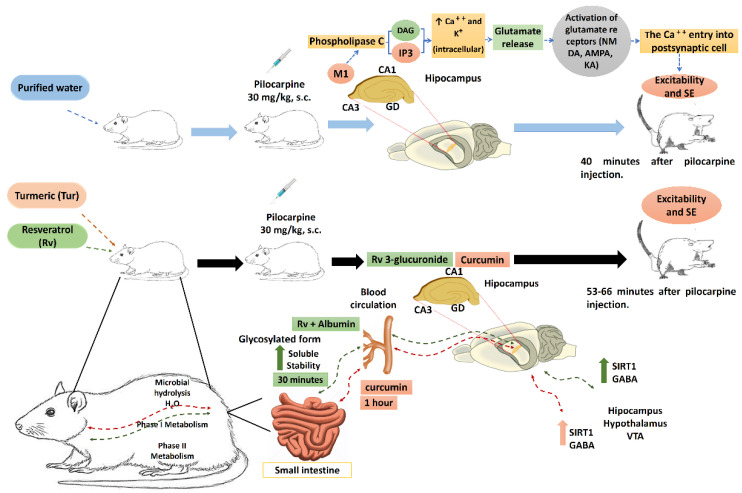
Theorical model of the protective mechanism of turmeric and resveratrol in status epilepticus (SE) induced with lithium–pilocarpine. Subcutaneous (s.c.), muscarine receptor 1 (M1), inositol triphosphate (IP3), diacylglycerol (DAG), dentate gyrus (GD), cornu ammonis (CA1 and CA3), ventral tegmental area (VTA), and sirtuin expression activator 1 (SIRT1). (Figure was prepared by the authors).

**Table 1 molecules-27-03835-t001:** Post-status epilepticus blood chemistry of rats treated with the vehicle, turmeric, resveratrol, or carbamazepine.

Analite	Vehicle (Purified Water)	Turmeric (150 mg/kg)	Turmeric (300 mg/kg)	Resveratrol (30 mg/kg)	Resveratrol (60 mg/kg)	Carbamazepine (300 g/kg)	Reference Intervals *
**Glucose**	8.43 ± 0.53	7.29 ± 3.30	7.00 ± 4.39	7.29 ± 3.30	8.43 ± 3.15	6.00 ± 0.00	6.00–10.00 mmol/L
**Urea**	18.43 ± 0.53	18.40 ± 0.76	16.99 ± 1.65	17.79 ± 0.80	17.71 ± 1.38	18.14 ± 0.89	10.70–20.0 mmol/L
**Creatinine**	22.86 ± 4.00	20.29 ± 5.05	20.71 ± 4.57	20.71 ± 5.34	18.14 ± 3.80	17.43 ± 2.63	11.00–28.0 µmol/L
**BUN**	6.22 ± 0.40	6.10 ± 0.33	6.10 ± 0.62	6.62 ± 0.07	6.47 ± 0.28	6.48 ± 0.10	3.00–7.00 mmol/L

Values are expressed as the mean ± SEM. * Source: Suckow et al., 2005 [14]; Giknis and Clifford, 2008 [15]; Sharp and Villano, 2013 [16]; Rivadeneyra et al., 2018 [17].

**Table 2 molecules-27-03835-t002:** Post-status epilepticus hepatic function tests of rats treated with the vehicle, turmeric, resveratrol, or carbamazepine.

Analite	Vehicle (Purified Water)	Turmeric (150 mg/kg)	Turmeric (300 mg/kg)	Resveratrol (30 mg/kg)	Resveratrol (60 mg/kg)	Carbamazepine (300 g/kg)	Reference Intervals *
**Total bilirubins**	0.18 ± 0.00	0.18 ± 0.05	0.18 ± 0.04	0.18 ± 0.05	0.18 ± 0.05	0.18 ± 0.052	0.04–0.20 mg/dL
**Direct bilirubin**	0.03 ± 0.00	0.03 ± 0.00	0.03 ± 0.00	0.03 ± 0.00	0.03 ± 0.00	0.03 ± 0.00	0.03–0.06 mg/dL
**Indirect bilirubin**	0.00 ± 0.00	0.00 ± 0.00	0.00 ± 0.00	0.02 ± 0.04	0.02 ± 0.04	0.04 ± 0.05	0–0.10 mg/dL
**AST**	143.40 ± 10.64	129.60 ± 13.02	142.3 ± 11.86	15.90 ± 5.52	137.70 ± 12.19	96.57 ± 21.95	63.00–157.00 UI/L
**ALT**	50.29 ± 1.25	43.14 ± 7.64	33.57 ± 10.21	46.14 ± 5.15	48.00 ± 2.94	43.43 ± 7.58	19.00–53.00 UI/L
**Alkaline phosphatase**	214.10 ± 77.29	135.4 ± 11.24	136.40 ± 23.29	186.70 ± 68.58	177.70 ± 29.75	146.4 ± 39.25	36.00–312.00 UI/L
**Total protein**	6.12 ± 0.40	6.10 ± 0.41	6.51 ± 0.45	6.27 ± 0.51	6.14 ± 0.19	6.58 ± 0.33	5.60–7.60 g/dL
**Albumin**	4.50 ± 0.36	4.57 ± 0.22	4.34 ± 0.35	4.55 ± 0.26	4.48 ± 0.41	4.25 ± 0.21	4.00–5.00 g/dL
**γ-Glutamyl-transferase**	15.74 ± 1.78	14.79 ± 1.43	14.86 ± 0.55	14.50 ± 1.83	17.86 ± 0.73	21.43 ± 0.78	8.8–24 UI/L

Values are expressed as the mean ± SEM. * Source: Suckow et al., 2005 [14]; Giknis and Clifford, 2008 [15]; Sharp and Villano, 2013 [16]; Rivadeneyra et al., 2018 [17].

**Table 3 molecules-27-03835-t003:** Post-status epilepticus complete blood cytometry of rats treated with the vehicle, turmeric, resveratrol, or carbamazepine.

Analite	Vehicle (Purified Water)	Turmeric (150 mg/kg)	Turmeric (300 mg/kg)	Resveratrol (30 mg/kg)	Resveratrol (60 mg/kg)	Carbamazepine (300 g/kg)	Reference Intervals *
**Erythrocytes**	8.05 ± 0.25	8.21 ± 0.21	8.36 ± 0.23	8.37 ± 0.22	8.21 ± 0.24	8.21 ± 0.25	7.8–8.65 mm^3^
**Hb**	14.61 ± 1.48	15.66 ± 1.16	15.90 ± 1.04	15.33 ± 1.16	15.93 ± 0.84	15.27 ± 0.66	13.20–17.10 g/dL
**Hto**	40.89 ± 1.95	41.44 ± 1.65	42.35 ± 0.83	42.29 ± 1.22	41.56 ± 2.18	42.26 ± 2.72	35–45%
**MCV**	54.39 ± 2.34	53.97 ± 1.64	54.02 ± 1.55	54.11 ± 2.52	55.06 ± 2.21	54.69 ± 1.92	45–65 fL
**MCH**	17.66 ± 0.96	17.79 ± 0.31	17.88 ± 0.16	17.57 ± 1.13	17.17 ± 0.52	17.49 ± 0.63	15.53–20.05 pg
**MCHC**	30.71 ± 0.48	30.53 ± 0.39	30.68 ± 0.35	28.30 ± 2.22	30.66 ± 0.59	30.29 ± 0.48	25.88–32.88 g/dL
**Leukocytes**	10.70 ± 5.88	9.18 ± 4.27	10.82 ± 4.58	10.73 ± 5.58	12.56 ± 3.97	9.50 ± 1.76	4.0–17.0 mm^3^
**Platelets**	782.70 ± 117.40	703.90 ± 107.80	862.70 ± 228.30	743.40 ± 220.10	629.70 ± 206.80	718.60 ± 168.10	300.0–1500.0 mm^3^

Values are expressed as the mean ± SEM. Rivadeneyra et al., 2017, 2018 [17,18]. * Source: Arcila et al., 2010 [19].

**Table 4 molecules-27-03835-t004:** Post-status epilepticus leukocyte differential count of rats treated with the vehicle, turmeric, resveratrol, or carbamazepine.

Analite	Vehicle (Purified Water)	Turmeric (150 mg/kg)	Turmeric (300 mg/kg)	Resveratrol (30 mg/kg)	Resveratrol (60 mg/kg)	Carbamazepine (300 g/kg)	Reference Intervals *
Segmented neutrophil	58.00 ± 2.87	59.71 ± 3.19	66.43 ± 1.71	54.43 ± 2.04	58.86 ± 2.77	58.00 ± 5.62	35–71%
Band neutrophil	0.57 ± 0.20	0.42 ± 0.20	0.28 ± 0.28	0.00 ± 0.00	0.00 ± 0.00	0.14 ± 0.14	0–2%
Lymphocytes	39.71 ± 2.37	35.43 ± 2.80	29.71 ± 2.39	43.57 ± 2.05	39.00 ± 2.92	35.71 ± 6.15	20–50%
Eosinophils	0.85 ± 0.40	1.42 ± 0.52	1.71 ± 0.71	0.71 ± 0.18	1.28 ± 0.28	1.00 ± 0.43	0–4%
Basophils	0.00 ± 0.00	0.42 ± 0.20	0.14 ± 0.14	0.14 ± 0.14	0.00 ± 0.00	0.85 ± 0.55	0–1%
Monocytes	0.85 ± 0.26	2.71 ± 0.77	1.71 ± 0.64	1.14 ± 0.26	0.85 ± 0.14	4.42 ± 1.08	0–5%

Values are expressed as then mean ± SEM. * Source: Arcila et al., 2010 [19].

## Data Availability

Not applicable.

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
