# Peer review of "Anticonvulsant Effect of Turmeric and Resveratrol in Lithium/Pilocarpine-Induced Status Epilepticus in Wistar Rats"

_molecules, 2022, doi:10.3390/molecules27123835_

Round 1

Reviewer 1 Report

This manuscript evaluated anticonvulsant effect by turmeric and resveratrol. Authors comprehensively evaluated effects using Wistar rats model with status epilepticus induced by lithium/pilocaropine. However, this manuscript lacks logic of conducting this research. Authors used turmeric and resveratrol for the evaluation. Resveratrol is pure chemical compound while turmeric is herbal medicine. So there is no reason to compare these two potential agent. Additionally, authors purchased turmeric from the market. Turmeric may come from various origins and depending on the harvest and processing condition, composition of herbal medicine may vary significantly. Usually, those pharmacological effect comes from those active components, so it is important to control amount of those active components while studying with herbal medicine. Therefore, it is risky to conclude by limited information of turmeric. It could have different result when tested with different origin of turmeric. Therefor, I cannot suggest this article to be published in this journal.

Here is few lists of errors that I found.

-Page 1, Line 31: what is full name of EE

-Page 1, Line 37: Purported à Should change the verb, definition of purported is appear or claim to be or do something, especially falsely.

-Page 2, Line 61: identify or discard à not usual phrase, rewrite this sentence.

-Page 2 figure 1 dosage of turmeric and resveratrol should be described, not just grouping 1 and 2. This is same for figure 2 and figure 3 as well.

Line 179 rewrite the sentence

-page, 9 4 Material and methods : numbering of subsection is redundant, please check.

Author Response

Please, see attachment file.

Reviewer 2 Report

  1. My major concern is that the influence of resveratrol on epileptic seizures has been assessed previously in several reports which are cited by the authors in the Discussion in lines 167-175. Please add new paragraph to the Discussion section explaining the rationale for assessing antiepileptic effects of resveratrol in this manuscript.

  1. Similarly, it has been previously shown in several publications that turmeric exerts antiepileptic effects. These reports are cited in the Discussion section (lines 157-166). Please add new paragraph to the Discussion section explaining the rationale for assessing antiepileptic effects of turmeric in this manuscript.

  1. The authors found that administration of turmeric and resveratrol did not deteriorate hepatic and renal function. This is not surprising because these compounds are often added to meals and exert many beneficial effects for human health.

What was the rationale for performing experiments assessing renal and hepatic function after administration of turmeric and resveratrol ? In other words, why did authors assume that tested compounds may deteriorate renal and hepatic function?

  1. Figure Captions are extremely short. More details should be included in Figure Captions.

  1. In all Figures please indicate doses of turmeric instead of turmeric 1 and 2. Please do the same with resveratrol.

  1. In Figure 2, why is the latency longer after application of the lower dose of turmeric compared to a higher dose? Lower dose should exert weaker antiepileptic effect (latency should be shorter).

  1. In Figure 4A why is the number of seizures higher after application of reservatrol 2 (higher dose) compared to reservatrol 1 (lower dose)? Higher dose should exert stronger antiepileptic effect (number of seizures should be lower). There is big difference between reservatrol 1 and 2 in Fig 4A. Please explain.

  1. Manuscript should be edited by a native speaker (see for example line 71, analysis of latency used to first generalize the seizures is incorrect English).

Author Response

Please, see attachment file.

Reviewer 3 Report

Manuscript ID: molecules-1717177

            The article with title “Anticonvulsant effect of turmeric and resveratrol in lithium/pilocarpine-induced status epilepticus in Wistar rats” by Eduardo Rivadeneyra-Domínguez et al discusses the effects of turmeric powder and resveratrol towards anticonvulsant property.

            This manuscript is well written with an appropriate introductory discussion and scientific soundness. This makes it valuable to the scientific community working in the field of natural products for status epilepticus and epilepsy treatment.

General comments-

Turmeric© capsule contained 350 mg of standardized turmeric. What would the curcumin content be in this capsule? As mentioned in the discussion, “In the model of epileptogenesis generated by unilateral intrahippocampal injection of 4 μg of kainate, rats that received 100 mg/kg/day of curcumin, the active ingredient in turmeric, had a reduction in seizures in 71.5% of the individuals [23]” Could the authors compare by means of actual curcumin content in your turmeric sample the dosage which directly caused the latency to seizures identified in their study? How was the solution of turmeric 150 mg/kg, turmeric 300 mg/kg prepared from the turmeric capsule. Please include in material and methods. Similarly for resveratrol capsule containing 500 mg of resveratrol, how was this solution prepared to be given to the animals?

Do resveratrol and turmeric components able to cross the Blood brain barrier (BBB), Please comment on metabolic fate or include some significant literature reference and discuss accordingly.

Since the blood chemistry, blood cytometry and hepatic function did not give significantly different results between groups, I am assuming metabolic fate of the products is an important aspect to consider discussing about. What do the authors feel?

Author Response

Please, see attachment file.

Round 2

Reviewer 1 Report

Authors have well addressed to reviewer's comment.

Still, there are a few grammatical errors. Please consult to native speakers for English correction.

Below are just few examples. There are still more errors which I have not mentioned.

Page 2 Line 60: "will be evaluate" is not grammatically right.

Page 7 Line 189~190: Please rephrase.

Reviewer 2 Report

I accept the manuscript in the present form.

The quality of writting should be improved. The manuscript should be edited by a naitive speaker.
